# Optical trapping of otoliths drives vestibular behaviours in larval zebrafish

Itia A. Favre-Bulle[1,2], Alexander B. Stilgoe [1], Halina Rubinsztein-Dunlop[1] & Ethan K. Scott [2,3]

The vestibular system, which detects gravity and motion, is crucial to survival, but the neural circuits processing vestibular information remain incompletely characterised. In part, this is because the movement needed to stimulate the vestibular system hampers traditional neuroscientific methods. Optical trapping uses focussed light to apply forces to targeted objects, typically ranging from nanometres to a few microns across. In principle, optical trapping of the otoliths (ear stones) could produce fictive vestibular stimuli in a stationary animal. Here we use optical trapping in vivo to manipulate 55-micron otoliths in larval zebrafish. Medial and lateral forces on the otoliths result in complementary corrective tail movements, and lateral forces on either otolith are sufficient to cause a rolling correction in both eyes. This confirms that optical trapping is sufficiently powerful and precise to move large objects in vivo, and sets the stage for the functional mapping of the resulting vestibular processing.

[1] School of Mathematics and Physics, The University of Queensland, St. Lucia, QLD 4072, Australia. [2] School of Biomedical Sciences, The University of Queensland, St. Lucia, QLD 4072, Australia. [3] Queensland Brain Institute, The University of Queensland, St. Lucia, QLD 4072, Australia. Correspondence and requests for materials should be addressed to H.R.-D. (email: halina@physics.uq.edu.au) or to E.K.S. (email: ethan.scott@uq.edu.au)

A cross vertebrates, the vestibular sensory organs comprise the otoliths, movements of which trigger hair cell activity to detect acceleration, and the semicircular canals, which are sensitive to rotational stimuli. In most vertebrates, vestibular processing involves the integration of information from these two structures into a coherent representation of the head's position and movement[1]. Larval zebrafish present a simplified version of this, in which the semicircular canals are not yet functional[2], and only the utricular otoliths detect vestibular stimuli (the saccular otoliths are involved with auditory perception)[3–5]. As such, the utricular otoliths, and the hair cells that they stimulate, represent a starting point from which all vestibular processing must originate[6]. This means that gaining physical control of the utricular otoliths would permit the exploration of the vestibular system in a stationary preparation, and would permit independent bilateral control of fictive vestibular stimuli.

Optical trapping (OT) is a well-established method for manipulating nanometre to micrometre-scale objects in complex media[7–10]. When laser light is highly focused, the intensity gradient near the focal point is large, and this gives rise to forces on transparent objects with different refractive indices to those of their surroundings. These forces enable the controlled confinement and movement of microscopic objects relative to their surroundings. Accordingly, OT has been used extensively to study the physical properties of microscopic objects (bacteria, nucleic acids, proteins and synthetic spherical particles) and their microscale environments[11–14]. Generally, OT exerts pN-range forces, and displacements as small as $5 \times 10^{-10}$ m have been

**Fig. 1** Optical properties of otoliths for OT. **a** Illustration of a focussed beam deflected by an irregular ellipsoid. **b** A wide-field image of a dissected otolith, showing internal irregularities (*Scale bar*, 10 μm). **c** Q factor (*blue curve*) for a scattered beam on a spherical particle calculated with ray optics approximations (see Supplementary Methods) along its *y*-axis (*dashed line, inset*). A positive (*upward*) force results from a trap at the bottom of the otolith (*inset*), and a negative (*downward*) force at the top. The effect is purely radial, with no predicted *x*-axis force (*yellow curve*). **d** Forces applied to a focussed laser by an otolith, measured with a PSD (see Supplementary Methods). *Red arrows* represent the direction and intensity (*arrow length*) of the forces at various positions. Contours show high intensity force regions in *yellow* and weaker intensity force regions in *blue*. **e** 3D representation of the total trap force across the otolith. The *black line* indicates the plane presented in the next panel. **f** The magnitude of the force in the *x*-direction (*yellow curve*) is weak across the *y*-axis. The *blue curve* shows the magnitude and direction of the force in the Y direction, and the *dashed black line* shows the magnitude of the total force applied by the otolith to the focussed beam. **g**–**i** Movements of a dissected utricular otolith under the influence of a 250 mW OT at its *right edge* (Supplementary Movie 1). Grid spacing is 100 μm

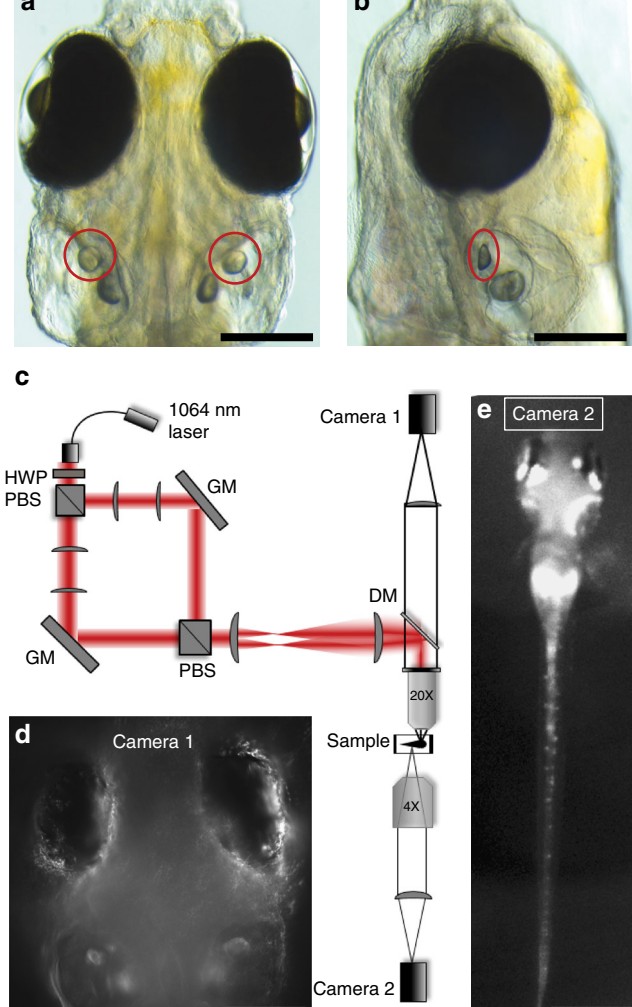

**Fig. 2** Optical setup for otolith OT and behavioural imaging. Dorsal (**a**) and lateral (**b**) views, indicating the location of the utricular otoliths (*circles*) in a 6dpf zebrafish larva (*scale bars*, 200 μm). **c** Experimental set up (see Supplementary Methods for details) for delivering a dual OT to the larva using a 1064 nm fibre laser, a half-wave plate (HWP), polarising beam splitters (PBS), gimbal-mounted mirrors (GM), a dichroic mirror (DM), and lenses to project the two traps into the sample via a 20X 1NA microscope objective. Camera 1 (**d**) allows targeting of the OTs and imaging of the eyes (*scale bar*, 200 μm), and camera 2 (**e**) permits imaging of tail movements (*scale bar*, 600 μm)

measured, but the exertion of stronger forces, and the movement of larger objects remains challenging. Furthermore, trapping objects in vivo, especially at depth, is difficult because of the scattering and power loss that result from passage through biological tissue[15]. To date, OT in vivo has been restricted to rather small objects (up to few microns) such as red blood cells[9], injected nanoparticles, erythrocytes, and macrophages[8].

Here, we perform an optical analysis of the otoliths in larval zebrafish, apply and measure OT forces to the otoliths using a focussed infrared laser beam, and characterise the relationships between perceived acceleration or rotation and the compensatory behavioural responses in the tail and eyes of the stationary zebrafish larvae. We show that the left and right ears make distinct and reciprocal unilateral contributions to postural adjustments of the tail, but that each ear is capable of driving both eyes as they compensate for perceived roll. Our results demonstrate that we

can apply controlled forces deep in intact tissues to large and irregularly shaped objects, such as the 55-micron otoliths, non-invasively in vivo. They also demonstrate the behavioural contributions made by each ear to perceived vestibular stimuli, and provide a stationary preparation for the elucidation of the underlying neural circuits.

## Results

**Optical properties of otoliths.** Because otoliths are relatively large and deep below the dorsal surface of the larva, OT of otoliths (Fig. 1a) presents particular challenges. Otoliths are composed primarily of crystalline calcium carbonate[16] and due to their aragonite structure they are birefringent with refractive indices of $n_\alpha = 1.53$, $n_\beta = 1.68$, $n_\gamma = 1.69$[17]. We find the utricular otoliths in 6-day postfertilisation (dpf) zebrafish larvae to be roughly 55 μm in diameter (Fig. 1b) and are located roughly 150 μm below the dorsal surface of the animal. On the basis of these characteristics and the known light scattering properties of biological tissue[15] (Supplementary Fig. 1 and Supplementary Methods), we modelled the nature of the forces that could be delivered in vivo. (Fig. 1c, Supplementary Fig. 2). These OT forces were calculated using ray optics methods (Fig. 1c, Supplementary Methods), assuming a spherical otolith. The predicted force is proportional to quality factor of the OT, Q, (Fig. 1c) and the laser power (Supplementary Methods), and its maximum (on the order of $5 \times 10^{-10}$ N for a 500 mW laser) occurs when the trap is at the sphere's edges. This modelling suggests that the focussed beam would have to be precisely positioned, as the force is predicted to drop by 20% with a shift of less than 2 μm away from the optimal position.

The actual forces cannot be measured experimentally in vivo, but since the otoliths are not perfect spherical crystals, it is important to measure and account for spatial variation in the refractive index across each otolith, which could result in changes in the locations and directions of maximum OT forces. We approached these measurements in vitro, using a light deflection method[18]. This involved scanning a tightly focussed beam across a surgically removed utricular otolith and measuring the average deflection of the scattered light (Supplementary Fig. 3 and Supplementary Methods). For each scanning position, we recorded the forces along the x-axis ($F_X$) and y-axis ($F_Y$) independently using a position sensitive detector, and the total force was calculated as the magnitude of the vector sum.

While the otoliths' structural heterogeneity led to some variability in the forces (Fig. 1d), we uniformly saw the strongest forces when the trap was at the otoliths' edges. The effects of birefringence were also evident around the otoliths' circumference, with two wide regions with stronger forces of 5.0 with standard deviation of ± 0.7 pN, and two narrower regions where the force was 3.0 ± 0.6 pN (Fig. 1d, e). Although this result proves that the birefringence has an effect on the total force exerted on the otoliths, the effect is moderate (comparable to that of the structural heterogeneity in the otolith). Relative to these effects, the trap force is much more dependent on the placement of the focal point near the edge of the otolith as suggested by our modelling (Fig. 1c). Moreover, OTs at the edges consistently produced forces in the radial direction in both the measurements and the model (Fig. 1c, d), and the variation of measured forces across the x- and y-axes generally agree well with modelling (*blue* and *yellow* curves in Fig. 1f vs. 1c) that did not incorporate factors such as heterogeneity and birefringence. As a final practical test of OT efficacy on otoliths, we performed trapping in vitro on surgically removed utricular otoliths. We found that a 1064 nm focussed laser beam can drag a free otolith using powers at or above 250 mW (Fig. 1g–i, Supplementary Movie 1).

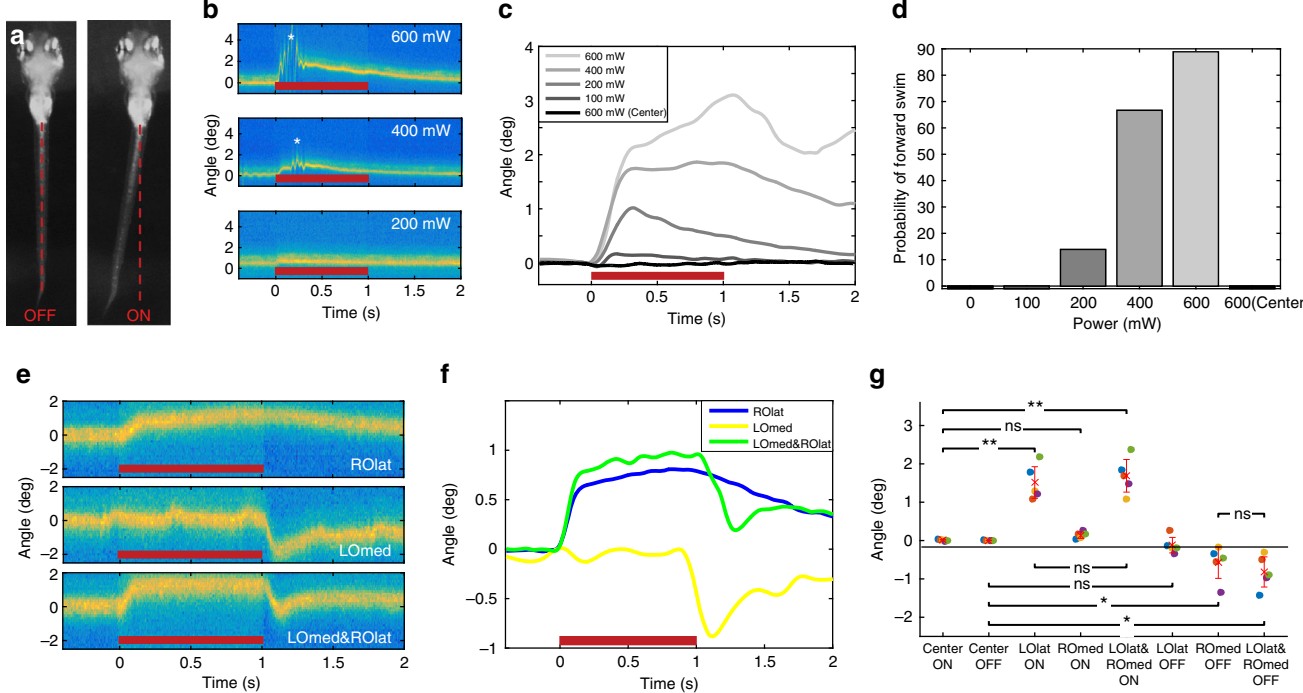

**Fig. 3** Otolith trapping results in coordinated compensatory movements in the tail. **a** Tail position of a larva before (*left*) and during (*right*) a 600 mW optical trap to the outside of the right otolith. **b** Tail positions during trap (*red bar*) with different laser powers. Forward swimming (*asterisks*) is evident during the 400 and 600 mW trials. **c** Tail deflection increases with laser power, and a 600 mW trap to the centre of the otolith has no effect (average of trials from one larva). Responses for all larvae are shown in Supplementary Fig. 4. **d** The probability of forward swimming increases with increased laser power ($n = 6$ larvae, 3 trials at each power). **e** Tail positions for a representative trial during a trap (600 mW) to the lateral edge of the right otolith (ROlat), the interior of the left otolith (LOmed), or a double trap of both. **f** Average of trials from one larva. The combined tail response is roughly a linear sum of the two separate traps' effects. Responses for all larvae are shown in Supplementary Fig. 5. **g** Responses to the onsets and offsets of different trap combinations. $n = 5$ larvae (*different colours*), and each point represents an average of 2–3 trials (after the application of exclusion criteria, see 'Methods' section). Mean +/− SEM is shown, *$p < 0.05$, **$p < 0.01$, paired $t$ test). Larvae were tested in both left/right orientations, but for clarity, LOlat/ROmed animals are flipped in this figure

**OT of otoliths results in compensatory vestibular behaviours.**
These otoliths are, to our knowledge, the most massive and most optically complex objects to be moved with OT. This trapping was, however, simplified by the fact that the otoliths were dissected out of the animal, and therefore easily targeted. Whether similar trapping will effectively apply forces in vivo, where the otoliths are deep within a complex milieu of tissues with varying refractive indices (Fig. 2a, b), is uncertain. Our modelling (Fig. 1c) suggests that these OT forces should be largely preserved within intact zebrafish larvae, but in vivo results are required to confirm this. It is also not assured that the OT forces placed on otoliths will be sufficiently physiological to trigger vestibular behaviours in the affected larvae. To address these questions, we designed a microscope for dual OT combined with two cameras for behavioural tracking of the larva's eyes and tail (Fig. 2c–e, see detailed description in Supplementary Methods). This setup permits us to apply two independently targeted OTs, one for each utricular otolith, through the dorsal side of the larva.

Using this setup, we performed OT on the otoliths of live 6dpf larvae in a tail-free immobilised preparation (see 'Methods' section). The orientation of the utricular otoliths, sitting flat in the ventral region of each ear (Fig. 2a, b) is the same as it was for the above modelling and in vitro trapping (Fig. 1). We found that a trap targeted to the lateral edge of one utricular otolith (the right otolith in the case of Fig. 3a) resulted in a deflection of the tail in the contralateral direction. As the OT was made more powerful, the magnitude of the tail bend also increased (Fig. 3b, c). Interestingly, powerful traps elicited both a strong tail deflection and oscillations representing a forward swimming motion

(Fig. 3b, d; Supplementary Movie 2). Supporting the idea that these are kinematically normal swim bouts[19–21], the pectoral fins also show oscillations during these movements (Supplementary Movie 2). These responses were not a function of heating, pain, damage to the otolith, or direct activation of hair cells from the IR irradiation[22, 23], as a laser targeted to the centre of the otolith (where it applies no coherent physical force) had no behavioural effect (Fig. 3c, d; Supplementary Movie 3). This demonstrates that OT can provide a fictive acceleration stimulus that feeds into behaviourally relevant circuitry.

We next tested the laterality of this behavioural response, and the contributions made by each of the two sides, by targeting the trap to the medial edge of one otolith and the lateral edge of the other. As demonstrated above, the lateral OT produced an outward force resulting in a contralateral bend of the tail as the trap was activated, and the tail gradually returned to its baseline position after the trap was turned off (Fig. 3e). An inward force on the opposite otolith had no effect on the tail position, but the tail made a bend in the opposite direction when the trap was turned off (Fig. 3e). Finally, we trapped both otoliths simultaneously, producing a bilaterally coherent fictive acceleration. This led to an active bend with stimulation from the OTs, and an active return to baseline when the traps were turned off (Fig. 3e, f; Supplementary Movie 4). The overall postural response to a bilaterally coherent stimulus appeared to be the linear combination of the two ears' independent contributions to the behaviour (Fig. 3f). Supporting this idea, a mathematical sum of the two independent responses closely mirrors the behavioural response to a dual trap (Supplementary Fig. 5), and a

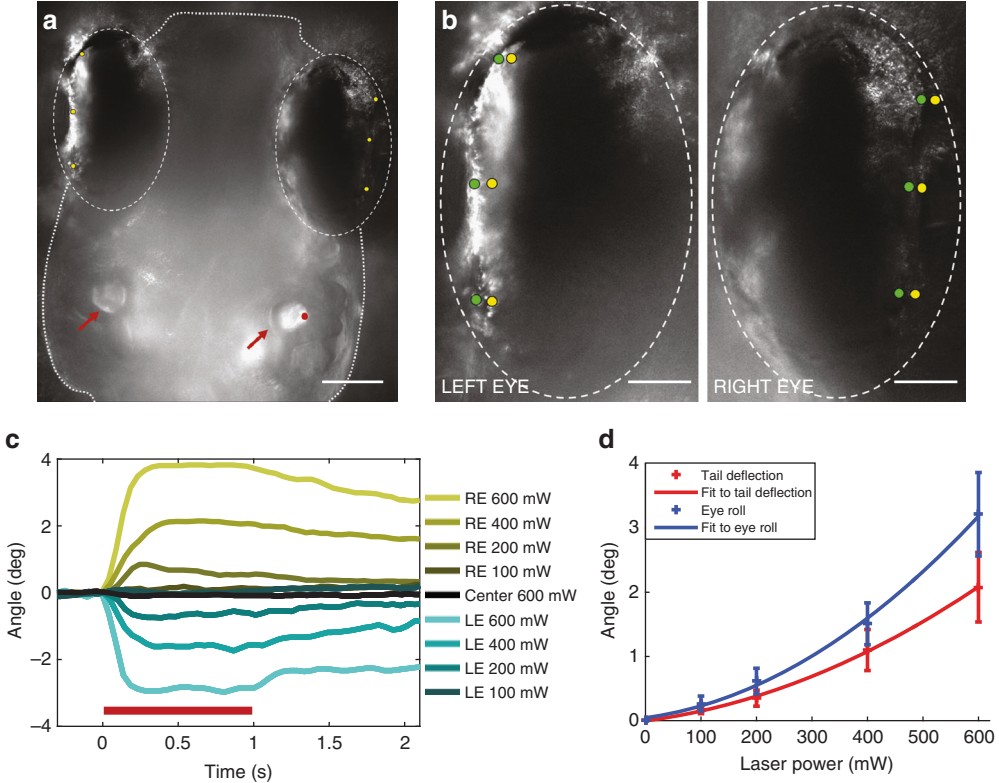

**Fig. 4** Otolith trapping results in coordinated compensatory movements in eyes. **a** Dorsal view of a larva. Otoliths (*red arrows*), the position of the trap on the lateral edge of the right otolith (*red dot*), and pigment landmarks on the eyes (*yellow dots*) are indicated. *Scale bar*, 100 μm. **b** Positions of pigment landmarks from **a** before (*yellow*) and during (*green*) a 600 mW trap. *Scale bar*, 50 μm. **c** Rotation of the left (*LE, blue*) and right (*RE, yellow*) eyes vs. the midline at a range of trap powers (average of three trials from one larva). Responses for all larvae are shown in Supplementary Fig. 6. **d** Average across all fish (*n* = 5) of the maximum deflection angle for the tail in *red*, and maximum eye roll in *blue*. Second order polynomial curves fitting these data are shown, and SEM is indicated for each power

quantitative analysis of the responses across five larvae (Fig. 3g) confirmed this.

Since real-world vestibular stimulation leads to stereotyped compensatory eye movements[24], we also tracked the positions of the eyes during OT (Fig. 4a). We found that outward forces to either otolith led to the rolling of both eyes in unison (Fig. 4b). No pitch nor yaw movement of the eyes was observed. As with the tail, the eyes moved more in response to stronger traps (Fig. 4c, d; Supplementary Movie 5), and OTs directed at the centres of otoliths produced no responses (Fig. 4c, Supplementary Movie 6). Inward forces did not affect the eyes, even in trials where they drove resetting movements in the tail.

## Discussion

It is unsurprising that these fictive vestibular stimuli result in both deflections of the tail and rolling movements of the eyes (Fig. 4d). Because it is more dense than its surroundings, an otolith (taking the left otolith as an example) would be drawn outward either by a linear acceleration of the animal to the right, or by gravity if the animal rolled to its left side. In the absence of corresponding visual or lateral line stimulation, and without functioning semi-circular canals, the motions would be indistinguishable to the larva's nervous system. The eye movements serve to stabilise the visual field in response to a perceived rolling motion, while the tail movements may be aimed at correcting for displacement, roll, or both. A detailed analysis of the tail kinematics, including possible torsional movements not detected in this study, would be necessary to gauge the effects that they would have in free-swimming larvae. The occurrence of forward swimming bouts during stronger fictive stimuli fits with the results of a recent

study showing that zebrafish larvae execute forward swimming as a means of stabilising their posture[25]. Just as free-swimming larvae perform a stabilising bout when they drift into a nose-down posture, our immobilised larvae perform swim bouts in response to stimuli simulating translation/roll. A summary of our fictive stimuli and accompanying behavioural responses are shown in Supplementary Fig. 7.

These results firmly establish that movements of the utricular otoliths alone are sufficient to drive compensatory responses across the body of a larval zebrafish. The tail movements resemble those resulting from optogenetic stimulation of the nucleus of the medial longitudinal fasciculus (nMLF) in larval zebrafish, shown by Thiele et al.[26]. In this study, activation of nMLF reticulospinal neurons led to ipsilateral deflections of the tail, with a gradual return to the centre after the activation was stopped, and forward swimming in response to powerful stimulation. Our similar observations suggest that vestibular signals from the ear are relayed through the nMLF to direct responses in the tail. Indeed, there are two categories of neuron located in the tangential nucleus (ascending and ascending/descending) that are well positioned to relay these signals from the utricular hair cells to the nMLF[27], thus subserving the tail responses that we observe. Our observations of the eyes represent the vestibulo-ocular reflex, a three-neuron circuit that has been described in various vertebrates, including larval zebrafish[27]. In this previous study, Bianco et al.[27] showed that changing a larva's pitch resulted in torsional eye movements, and that each otolith contributed roughly equally to the rotation of both eyes. Furthermore, removal of the otoliths caused changes in the eyes' vertical position, revealing their control over rolling eye movements like those described in this

study. In mapping the anatomy and function of the neurons carrying vestibular information from the utricular hair cells, these prior studies provide a framework for the circuitry that bridges our OT of the otoliths to the behavioural responses that we observe.

From a technical perspective, our results demonstrate the tractability of OT, even of large objects deep within intact, behaving animals. While biological tissues present the challenges of light scattering and power loss[15], we were nonetheless able to use OT effectively with standard microscope components and a laser of moderate power. This confirmed the results of our detailed in vitro analysis of the otoliths' properties as an OT target. The size and optical complexity of the otoliths, combined with their depth in the larvae, suggest that this method should be easily adapted to other in vivo OT applications, especially in small model systems. Specifically related to this system, OT removes the physical movement from experiments of an inherently motion-detecting modality. This provides a stationary imaging platform in which to observe neural circuit activity during controlled, sustained, and bilaterally regulated vestibular stimuli.

## Methods

**Animals.** All procedures were performed with approval from The University of Queensland Animal Welfare Unit (in accordance with approval SBMS/305/13/ARC). Zebrafish (*Danio rerio*) larvae were maintained at 28.5 °C on a 14 h ON/10 h OFF light cycle. Adult fish were maintained, fed, and mated as previously described[28]. All experiments were carried out in *nacre* mutant larvae of the Tupfel long fin strain[29].

**Sample preparation.** 6 dpf larvae of the Tupfel long fin strain were immobilised dorsal side up in 2% low melting point agarose (Progen Biosciences, Australia) on microscope slides. The agarose surrounding the tail was freed by removing segments of agarose perpendicular to the tail until reaching the swimming bladder. Larvae were then transferred to the imaging room and allowed to acclimate for 15 min prior to imaging on the custom-built dual OT microscope presented in Fig. 2.

**Behavioural experimental protocol.** The tail deflection and eye roll were studied with a range of OT powers (50, 100, 200, 400, and 600 mW) on the lateral side of one otolith, and 600 mW in the centre of that same otolith. The tail deflection was also studied with three different trapping conditions: 600 mW on the lateral side of one otolith, the medial side of the other otolith, and both traps simultaneously.

Each trial was repeated three times with 1 s exposure time and 9 s waiting time between trials. The different combinations of OTs were presented to the animal in random order. Since all animals received the same fictive stimuli, randomisation and experimental blinding were not used.

**Behavioural exclusion criteria.** Every trapping condition was presented in three separate trials. To ensure that spontaneous off-target behaviours were not being included among our data, we excluded trials in which:

- Spontaneous swimming occurred less than 1 s before OT initiation,
- Spontaneous swimming occurred less than 1 s after OT termination, or
- Escape behaviour occurred during or within 1 s of the OT.

Spontaneous swimming (rapid oscillations of the tail from side to side) was readily distinguished from postural changes occurring during the experiments, and only the former were used as a rationale for exclusion. All animals were given three trials for each stimulus (trap location and power), and for all animals and all stimuli, either one or zero trials were excluded. The means that the values shown in Fig. 3 represent averages of either two or three trials for each condition.

**General.** See Supplementary Methods for details on the optical setup, tail and eye tracking and modelling of forces in vivo with Monte Carlo and ray optics model. Code for tail tracking and Monte Carlo analysis can be accessed through the corresponding authors. See Supplementary Movies 1–6 for illustration of behavioural responses. The Supplementary Methods contain additional experimental details.

**Code availability.** The code used in this study is available from the corresponding author on request.

**Data availability.** The data that support the findings of this study are available from the corresponding author upon reasonable request.

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

## Acknowledgements

We thank Anatoli Kashchuk for producing animations and members of the Scott and Rubinsztein-Dunlop labs for feedback on the manuscript. Support was provided by the

an NHMRC Project Grant (APP1066887), ARC Future Fellowship (FT110100887), a Simons Foundation Explorer Award (336331), and two ARC Discovery Project Grants (DP140102036 & DP110103612) to E.K.S.; an ARC Discovery Project (DP140100753) to H.R.-D.; and a UQ Postgraduate Scholarship to I.A.F.-B.

## Author contributions

I.A.F.-B., E.K.S and H.R.-D.: Conceived the project. I.A.F.-B. and A.B.S.: Performed the otolith scanning experiments and analyses. I.A.F.-B.: Performed the behavioural experiments, Monte Carlo model and analyses. I.A.F.-B., A.B.S., E.K.S. and H.R.-D.: Wrote the manuscript.

## Additional information

**Competing interests:** The authors declare no competing financial interests.

