## [Peer Review File · Nature Communications]

Reviewers' comments:

Reviewer #1, an expert in optical trapping (Remarks to the Author):

The manuscript by Favre-bulle et al. describes a study where optical tweezers are used to exert forces on otoliths in living zebrafish. They show that slight displacements of the otoliths cause tail and eye movement. It is highly exciting that the authors here demonstrate external control of zebrafish movement by a focused laser beam. Also, this paper opens an avenue of exciting studies regarding the mechanisms governing motility and regarding how nerve conduction network guides motion. It is an elegant way to address these questions. However, the authors need to consider the following issues.

Major issues:

1. Figure 1 regards optical trapping in vitro of an otolith. The forces here calculated have essentially no relevance for the in vivo investigations since the environment is entirely different in vivo. The tissue causes aberration which will diminish the forces in some unknown manner and the landscape inside the fish is visco-elastic (not purely viscous). Hence, entirely different calibration procedures are needed to estimate forces in vivo. Therefore, I suggest the authors move Fig 1 to the supplementary information. Also, in lines 68-70 the authors need to explicitly state that this modeling only is of any relevance in vitro (not inside the fish).
2. In general, I urge the authors to significantly decrease the in vitro descriptions of the trapping of the otolith and instead spend more space of the manuscript text on their exciting in vivo results. For instance, the equation in line 100 is trivial and should be removed.
3. How deep inside the zebrafish are the otoliths located? Compared to the study by Johansen et al (ref 7) where trapping is demonstrated inside living zebrafish, do the authors of the current manuscript go deeper into the fish?
4. What is the lateral displacement of the otolith in vivo? Please plot the displacement as a function of laser power. (Probably the otolith displacement can be deduced from the videos).
5. How does this displacement distance relate to the deflection angle for the kinocilium and hair cells to depolarize hair cells and initiate signal transduction?
6. An important and necessary control is inactivate the hair cell response and test whether the OT can still cause a deflection.
7. Another interesting control is to focus the OT next to the otolith and test whether the laser itself can elicit signal transduction from the hair cells.
8. In lines 154-157 the authors state the interesting hypothesis that the overall response to a coherent stimulus appears to be a linear combination of the two ears' independent contributions. To test and potentially prove this hypothesis the authors should draw the mathematical sum of the two contributions as well as the measured total contribution in Fig 3f and on Extended Data Figure 2.
9. Figure 3 conveys the most interesting results of the paper and it is very dense. Therefore, I suggest that Figure 3 is split into 2 separate figures: One figure presenting the tail results and another figure presenting the eye results. (There should be room for this if Fig 1 is moved to the supplements). As Fig 3 is currently presented, it is impossible to read, e.g., Fig 3g.
10. The 'exclusion criteria' briefly mentioned in the caption of Fig 3 and detailed in the supl info (lines 89-96) raise some concern. It is crucial that exclusion criteria do not impose a bias on the results. 'Spontaneous swimming occurred less than one second after OT termination': With this criteria there is a risk that only curves (as, e.g., shown in Extended Data fig 1) where a decrease in angle after irradiation are chosen, hence, it is a selected result. And then I do not understand lines 95-96: Over the three trials of every trapping condition, either two or three passed the exclusion tests... What does this mean? Which 3 trials? For instance in Extended Data Fig 3 five different fish are presented. Has each fish been tested 3 times?
11. Extended Data Figure 4: How many times were these experiments repeated? (n=?). What could be a biological explanation of the observation in (i)? Is it statistically significant?
12. Concerning the eye movement, I suggest the authors quantify the movement of the tracer

spots in the eyes upon laser irradiation, plot this displacement as a function of laser power. It would also be relevant to plot the corresponding tail deflection as a function of laser power and see if the two responses, eye and tail, are linearly related (or if a non-linear neuronal response is at play).

Minor issues:

13. Fig 1f, the dashed black line shows the MAGNITUDE of the total force (not just the 'total force' along some direction).

14. In line 214 the authors write that trapping was performed with a low numerical aperture. The only value I could find for the numerical aperture was in the suppl info line 123 (describing the in vitro results). Here, the NA was 1.33 which I would not call 'low'. Please state which NA was used for the in vivo results.

15. The laser powers stated (e.g., in Fig 3), where are they measured?

16. In the movies it appears that not only the tail, but also the little fins move in a manner that correlate to the laser power. I suggest that the authors also quantify this fin behavior or at least discuss it.

If all above issues are dealt with in a satisfactory manner, I judge the manuscript appropriate for publication in Nature Communications.

Reviewer #2, an expert in zebrafish neurobiology and vestibular processing (Remarks to the Author):

In this manuscript, Favre-bulle and colleagues report a methodological advance: optical trapping of otoliths. The work begins with a theoretical treatment and movement of dissociated otoliths. Next, the authors use their apparatus to move the otoliths in vivo in the larval zebrafish, measuring the tail and eye movements that follow displacement. The authors are fortuitous in that the well-understood circuitry of the vestibular system has been exhaustively characterized in a number of vertebrate species, and thus their findings are straightforward to interpret. Subjectively, this work establishes a useful tool that enables a number of different experiments, and extends our understanding of the mechanisms by which animals achieve stable posture. As such, I find the work to be creative, original, and I expect this paper will be of considerable interest to a number of different communities from optical physicists to neurophysiologists. The analyses are adequate to support the claims made, and the data are convincing. Below are a number of concerns and suggestions, which can generally be addressed by textually.

1. The authors frame a number of findings as novel in the second-to-last paragraph of the manuscript that Bianco et. al. 2012 speaks to, directly and indirectly. Specifically, Bianco et. al. show that the tangential nucleus neurons relay vestibular information from the otoliths to the neurons in the nucleus of the MLF (Fig 4 & 5). The authors remark on the similarity to the Thiele findings, but they would strengthen their case by noting that the anatomical basis for such a link in the fish is well-established.

2. Similarly, the authors say "In this previous study [Bianco et. al.], changing a larva's pitch resulted in torsional eye movements, while we see rolling eye movements in response to a perceived roll of the larva's body. This underscores the flexibility of the VOR in larval zebrafish...." Bianco et. al. characterized the roll VOR extensively (Figure 3), and demonstrated by unilateral and bilateral utricular removal that each eye's movement is comprised of inputs from both utricles. The data here thus *confirms* the results in Bianco et. al.; the choice of "...while we see..." [207-208] suggests more. Further, there's nothing particularly "flexible" or "simple" [209-210] about the gravito-inertial VOR in the larval zebrafish. It works (and, to a first approximation, is wired) precisely the same way as in every other vertebrate.

3. The third-to-last paragraph (187-196) appears to confuse the tilt-translation ambiguity (the inability of a single inertial detector to disambiguate linear acceleration and rotation) with the basics of reflexive stabilization of posture and gaze. The VOR will happen whether the fish is tilted or linearly translated, as (presumably) would the tail movements. These two are not “unrelated:” on the contrary, evidence from other vertebrates suggest that they represent a collective attempt to correct destabilizations in both posture and gaze. As fish maintain a dorsal-up posture, if a fish were to roll (or be translated in the dark) to the left, it would be expected counter-roll its eyes to the right. Similarly, it ought do whatever it needed to its tail to ensure that it rolled to the right (more on this below). These have nothing to do with the nature of the stimulus that displaces the otolith. This confusion is also present in [33] where the authors separate acceleration from roll. There is no evidence that the fish perceives the two differentially, and the fact that the authors observe eye movements and tail movements do not disambiguate the two.

4. The authors claim that the tail movements would “likely represent postural changes.” [193] I encourage the authors to expand on this. Specifically, the authors should offer a compelling physical basis for this interpretation. I encourage the authors to look at Ehrlich and Schoppik, 2017. That paper found that any forward translation was sufficient to correct pitch. There is no reason to suppose that this finding wouldn’t extend to roll tilts as well. Specifically, if a fish translates, as it does when it bouts, its torpedo-like morphology is inherently stabilizing. Such an interpretation is consistent with the supplemental video and in (Figure 3d), where stronger stimuli are accompanied by an increased probability of forward swims. N.B. I may have missed it but I don’t believe the authors explicitly describe what constitutes a “forward swim,” though I surmise it is what we see in the caudal tail in the video. If the authors really want to propose that it is the tail angle and not the instantiation of the swimming that would stabilize posture I strongly encourage them to propose a testable mechanism by which a yaw angle of the tail would serve to stabilize the roll axis of the fish. They might model the effects of a body angle given the constant angular acceleration in pitch to which larvae are subject (Ehrlich and Schoppik 2017); the combined pitch & yaw might be sufficient to roll the fish properly for small deviations from dorsal-up. Similarly, it would be good to know if the first deflection of the tail in a forward swim was oriented preferentially to the left/right depending on the otolith stimulus that generated it? Slight deviations in the strength of the beat might be similarly useful in stabilizing posture. Ultimately, larval zebrafish morphology is sufficient to ensure that simply generating a swim bout would stabilize roll, regardless of direction.

5. Analysis of the eye movement data, while generally adequate for the claims that the authors make, is difficult to interpret given the authors’ choice of example movies. Specifically, the fish make spontaneous naso-temporal saccades and relax the eyes along this axis; this appears to be visible in the three-part video, particularly in the third part. If that particular video is comprised of three videos put together (as I’m guessing because of the displacements of the eye and the fish body), it would help to insert blank frames so the readers are not misled into thinking it contiguous. The authors would also do well to analyze any rostro-caudal changes in the pigment fiducials, and not just medio-lateral. A truly medio-lateral deflection of the utricle would produce no systematic rostro-caudal changes that would be the hallmark of a torsional (i.e. response to pitch tilts) eye movement. I note this because the authors claim that [161] “OTs directed at the centres of the otoliths produced no responses (Fig 3j Movie S6) but repeated viewings of Movie S6 shows a clearly discernible movement of the fish’s right eye. Finally, while I’m loathe to suggest new experiments, there are clear predictions for the eye movements that force on the rostral or caudal part of the utricle should produce, not just the medial/lateral edge. The manuscript would be considerably stronger for the characterization of eye and tail responses to the other two directions to which the utricle is sensitive.

6. The authors show the utricle moving on order 100um in Figure 1. I was surprised not to be able to see any deflection at all in Movies S5 and S6. Presumably this is because the otolith itself is tethered, but the authors might comment on this (unless I missed it somewhere)

7. The authors rely heavily on the "centre" directed stimulus to control for "heating, pain or damage" to the otolith. I note, though, that there is an extensive literature on laser-induced currents in hair cells: Rabbitt RD et. al. 2016, Rajguru 2011, and for shorter wavelength NIR stim Xia et. al. 2014 10.1109/EMBC.2014.6944163 . The manuscript would be stronger for a discussion of these results, as the "centre" stimulus does not adequately control for the possibility that the laser light itself could be sufficient to excite utricular hair cells, as shown by a number of other groups.

[40] should read "sensitive *to* rotational"

“Optical Trapping of Otoliths Drives Vestibular Behaviours in Larval Zebrafish”
Manuscript NCOMMS-17-01858-T
Response to Reviewers

The authors thank the editor and two reviewers for their assessments and comments for improvement of this manuscript. The reviewers’ complete and unedited comments are shown below in italics, and our responses to these comments are intercalated. Changes made to the manuscript in response to the comments are tracked in the .docx file that we have uploaded to the online submission system.

Reviewer #1, an expert in optical trapping (Remarks to the Author):

The manuscript by Favre-bulle et al. describes a study where optical tweezers are used to exert forces on otoliths in living zebrafish. They show that slight displacements of the otoliths cause tail and eye movement. It is highly exciting that the authors here demonstrate external control of zebrafish movement by a focused laser beam. Also, this paper opens an avenue of exciting studies regarding the mechanisms governing motility and regarding how nerve conduction network guides motion. It is an elegant way to address these questions. However, the authors need to consider the following issues.

Major issues:

1. Figure 1 regards optical trapping in vitro of an otolith. The forces here calculated have essentially no relevance for the in vivo investigations since the environment is entirely different in vivo. The tissue causes aberration which will diminish the forces in some unknown manner and the landscape inside the fish is visco-elastic (not purely viscous). Hence, entirely different calibration procedures are needed to estimate forces in vivo. Therefore, I suggest the authors move Fig 1 to the supplementary information. Also, in lines 68-70 the authors need to explicitly state that this modeling only is of any relevance in vitro (not inside the fish).

This is an important point. Including a detailed description (in Figure 1) related to the optical physical properties of the otolith is essential as we can show that the force is radial and also that the location of the maximum forces is highly localized. As can be seen from this analysis, the force is much greater at the edges of the otolith and goes almost to zero at the center. We feel that this in vitro study is illustrative to what the direction and strength of the forces is within the otolith. The study also shows that we can map the forces acting throughout the otolith giving us the distribution dependence on the structure. As the estimation of force is related to the maximum force that can be exerted the forces that are present in the zebrafish can only be smaller than that force.

Furthermore, although the strength of the trap in vitro has little predictive power for the strength of the trap in vivo (as the reviewer correctly points out), certain optical properties of the otolith for optical trapping (the shape of the curves in 1c-f, the otoliths’ birefringence, and the ways in which trap strength changes with subtle movements of the laser) will be preserved in vivo, and are important considerations for our subsequent experiments. Bearing all of this in mind, we would request to leave Figure 1 in the main

text.

In addressing the reviewer's point about unknown effects of biological tissue on the strength of the trap, we referred in the main text to our earlier work (Favre-Bulle et al, Scientific Reports) relating to the aberrations of the laser beam going through the tissue and we have run the simulations for the current situation and the results have been added to the main text (Fig. 1c and Lines 69-78) and the details of the calculation has also been described in the Supplementary section.

2. In general, I urge the authors to significantly decrease the in vitro descriptions of the trapping of the otolith and instead spend more space of the manuscript text on their exciting in vivo results. For instance, the equation in line 100 is trivial and should be removed.

Agreed. We have removed this equation and replaced it with a simple statement that delivers the same message (Lines 105-106). Based on the same rationale that we advanced for the last question, we have left other details of the physical analysis in place.

3. How deep inside the zebrafish are the otoliths located? Compared to the study by Johansen et al (ref 7) where trapping is demonstrated inside living zebrafish, do the authors of the current manuscript go deeper into the fish?

The Johansen study reported trapping at depths on the order of 50 μm (the exact figure is not clearly stated), and the otoliths are roughly 150 μm below the surface of the animal. So yes, we have gone deeper. However we are not trapping the otolith in the true sense of capturing it to the beam, we are merely applying forces to it using OT. Due to the uncertainty about the prior study's exact limits in terms of depth and the different use of OT, we have resisted making direct performance comparisons across the studies in our manuscript.

4. What is the lateral displacement of the otolith in vivo? Please plot the displacement as a function of laser power. (Probably the otolith displacement can be deducted from the videos).

Because the otoliths are tightly tethered within the ear, their movement is undetectable at the magnification that we are using for this study. From the videos in the supplementary section it is evident that that we cannot observe the movement of the otolith; we presume that it is less than one pixel. This makes sense, given that the movements must be within the range that hair cells' kinocilia can tolerate as they flex. We have done a few relevant exploratory studies that are not in the manuscript. In one, we can detect movements of the otoliths in vivo in 1 and 3 day postfertilization (dpf) embryos, but this is when the otoliths are much less tethered, and before the vestibular system is likely to be functional. We have done a small amount of in vivo imaging in 6dpf larvae using a 100X imaging

objective, and have (in some cases) observed very small (one or a few pixels) movements of the otoliths in the expected direction.

5. How does this displacement distance relate to the deflection angle for the kinocilium and hair cells to depolarize hair cells and initiate signal transduction?

As noted above, the very small distances involved make it difficult to address this question.

6. An important and necessary control is inactivate the hair cell response and test whether the OT can still cause a deflection.

Given the absence of a response to our centre trap, and the reciprocal effects that we see to traps on the medial and lateral sides of the otolith, we can think of no other plausible mechanisms for the transduction of this signal. The only object being manipulated directly is the otolith, and the hair cells are firmly established as the only conduit of information from the otolith into the nervous system. That the effects are reciprocal for medial and lateral traps shows that this is not an off-target stimulation of the hair cells themselves or any downstream circuitry. We feel that this conclusion is well supported by our data. We are also concerned about the technical challenges and confounds to interpretation that hair cell inactivation present. Optogenetic silencing is difficult to perform against numerous neurons specifically and simultaneously, and selective laser ablations of this number of cells would create so much damage in the area that the movement of the otoliths or the function of downstream circuitry would be called into question. We have not made changes in response to this comment in the revised manuscript, but we are happy to discuss this issue further if the editor or reviewer are not convinced.

7. Another interesting control is to focus the OT next to the otolith and test whether the laser itself can elicit signal transduction from the hair cells.

We have done this experiment, and do not see behavioral responses. We do not feel that this is an easily interpreted experiment, though. A positive response could suggest that heat or damage is responsible for the effect, or that convection in the ear fluid is moving the hair cells. A failure to respond suggests that it is the force placed on the otolith that is responsible for the behavior. Overall, we feel that the trap at the center of the otolith is a better control, as the presence or absence of a physical force on the otolith is the only variable between the two treatments. If the reviewer feels strongly, we can report our experiments from outside of the otolith as a way of addressing comments 6 and 7.

8. In lines 154-157 the authors state the interesting hypothesis that the overall response to a coherent stimulus appears to be a linear combination of the two ears' independent contributions. To test and potentially prove this hypothesis the authors should draw the

mathematical sum of the two contributions as well as the measured total contribution in Fig 3f and on Extended Data Figure 2.

Thank you for this suggestion. In response, we have added material to Supplementary Fig. 5 (previously Extended Data Figure 2). For the responses averaged across all fish, we have calculated a mathematical sum of the two responses through time, and added the resulting curve to the figure. The result closely mirrors the actual response from the fish during a dual trap. The changes can be found in the final panel of Supplementary Fig. 5, and a reference has been added to the main text (Line 175-178), where we also note the quantitative analysis in Fig 3g supporting this hypothesis.

9. Figure 3 conveys the most interesting results of the paper and it is very dense. Therefore, I suggest that Figure 3 is split into 2 separate figures: One figure presenting the tail results and another figure presenting the eye results. (There should be room for this if Fig 1 is moved to the supplements). As Fig 3 is currently presented, it is impossible to read, e.g., Fig 3g.

This is a good suggestion. Thank you. We have split the eye data out of Fig. 3, and have created a brief Fig. 4 to accommodate them.

10. The 'exclusion criteria' briefly mentioned in the caption of Fig 3 and detailed in the suppl info (lines 89-96) raise some concern. It is crucial that exclusion criteria do not impose a bias on the results. 'Spontaneous swimming occurred less than one second after OT termination': With this criteria there is a risk that only curves (as, e.g., shown in Extended Data fig 1) where a decrease in angle after irradiation are chosen, hence, it is a selected result. And then I do not understand lines 95-96: Over the three trials of every trapping condition, either two or three passed the exclusion tests... What does this mean? Which 3 trials? For instance in Extended Data Fig 3 five different fish are presented. Has each fish been tested 3 times?

Our apologies. This should have been clearer in our initial submission. Spontaneous swimming takes the form of rapid bilateral alternations of the tail position, and these would obfuscate any postural changes occurring at the same time. Such swim bouts are easily identified, and can be clearly differentiated from further positive postural changes. The latter was not used as a rationale for excluding a trial.

Yes. Each animal was given three trials of each stimulus. Since, at most, one trial was excluded in each condition, the resulting data represent the averages of either two or three trials. We have rewritten Methods section (Behavioural exclusion criteria) to clarify these issues. Our changes are tracked on this revised document.

11. Extended Data Figure 4: How many times were these experiments repeated? (n=?). What could be a biological explanation of the observation in (i)? Is it statistically

significant?

Supplementary Fig. 7 (previously ED Fig 4) is intended to be a summary figure. It reflects data shown previously in the manuscript, specifically in Figure 3 (and Figure 4 in the revised version). Panel (i) illustrates the contralateral bend that is seen at the cessation of a medial optical trap, as reported in panels a, b, and c of the new Fig 4. Its statistical significance is reported in new Fig 4c. It is underpinned by data from 5 larvae (Supplementary Fig. 4), with 2-3 trials per stimulus in each larva. We have added a sentence to the legend for Supplementary Fig. 7 to clarify this (Lines 156-159).

12. Concerning the eye movement, I suggest the authors quantify the movement of the tracer spots in the eyes upon laser irradiation, plot this displacement as a function of laser power. It would also be relevant to plot the corresponding tail deflection as a function of laser power and see if the two responses, eye and tail, are linearly related (or if a non-linear neuronal response is at play).

It is indeed an interesting point, thank you. We added the suggested data into Figure 4.

Minor issues:

13. Fig 1f, the dashed black line shows the MAGNITUDE of the total force (not just the 'total force' along some direction).

Thank you. We have changed the text to correct this inaccuracy (Lines 91-93).

14. In line 214 the authors write that trapping was performed with a low numerical aperture. The only value I could find for the numerical aperture was in the suppl info line 123 (describing the in vitro results). Here, the NA was 1.33 which I would not call 'low'. Please state which NA was used for the in vivo results.

True. We have adjusted this simply to say that the trapping was accomplished with "standard microscope components" (Line 256). While this is not specific, it serves the purpose of this sentence in a summary paragraph: to bring the experimental results into a context where other scientists can use it.

15. The laser powers stated (e.g., in Fig 3), where are they measured?

These powers were measured using a power meter at the same height as the position of the otolith. This information has been added in the Supplementary Methods (Lines 108-111).

16. In the movies it appears that not only the tail, but also the little fins move in a manner that correlate to the laser power. I suggest that the authors also quantify this fin behavior or at least discuss it.

Interesting. This is expected as part of a coordinated swim behavior, but we had not noted it until following up on this comment. In most trials, this movement is blocked because the pectoral fins are caught in the agarose encasing the head. It provides further evidence that these behaviours, seen with high power traps, are kinematically normal forward swim bouts. We have added mention of this, along with a relevant reference (Lines 157-158).

If all above issues are dealt with in a satisfactory manner, I judge the manuscript appropriate for publication in Nature Communications.

Reviewer #2, an expert in zebrafish neurobiology and vestibular processing (Remarks to the Author):

In this manuscript, Favre-bulle and colleagues report a methodological advance: optical trapping of otoliths. The work begins with a theoretical treatment and movement of dissociated otoliths. Next, the authors use their apparatus to move the otoliths in vivo in the larval zebrafish, measuring the tail and eye movements that follow displacement. The authors are fortuitous in that the well-understood circuitry of the vestibular system has been exhaustively characterized in a number of vertebrate species, and thus their findings are straightforward to interpret. Subjectively, this work establishes a useful tool that enables a number of different experiments, and extends our understanding of the mechanisms by which animals achieve stable posture. As such, I find the work to be creative, original, and I expect this paper will be of considerable interest to a number of different communities from optical physicists to neurophysiologists. The analyses are adequate to support the claims made, and the data are convincing. Below are a number of concerns and suggestions, which can generally be addressed by textually.

1. The authors frame a number of findings as novel in the second-to-last paragraph of the manuscript that Bianco et. al. 2012 speaks to, directly and indirectly. Specifically, Bianco et. al. show that the tangential nucleus neurons relay vestibular information from the otoliths to the neurons in the nucleus of the MLF (Fig 4 & 5). The authors remark on the similarity to the Thiele findings, but they would strengthen their case by noting that the anatomical basis for such a link in the fish is well-established.

Thank you for pointing out this omission. We have added a sentence noting this anatomical connection, and noting how it completes the circuit that we are presumably triggering (Lines 240-243).

2. Similarly, the authors say “In this previous study [Bianco et. al.], changing a larva’s pitch resulted in torsional eye movements, while we see rolling eye movements in response to a perceived roll of the larva’s body. This underscores the flexibility of the VOR in larval zebrafish....” Bianco et. al. characterized the roll VOR extensively (Figure 3), and demonstrated by unilateral and bilateral utricular removal that each eye’s movement is comprised of inputs from both utricles. The data here thus *confirms* the results in Bianco et. al.; the choice of “...while we see...” [207-208] suggests more. Further, there’s nothing particularly “flexible” or “simple” [209-210] about the gravito-inertial VOR in the larval zebrafish. It works (and, to a first approximation, is wired) precisely the same way as in every other vertebrate.

Our reading of Bianco et al is that the gain measurements with otolith removal (Figure 3A-D) are for pitch-tilt stimuli and torsional rotation of the eyes. We were cautious of over-comparing our results to Bianco’s because both utricular otoliths experience the same rotation during pitch changes, while the otoliths experience reverse rotations (relative to the midline) during a roll. The eye roll measurements in Fig 3E and F of Bianco are in response to the removal of the otoliths, rather than rotational stimuli. Here, the opposing effects seen from removing the left and right otoliths is in line with our results. The relevant paragraph has been rewritten to address the reviewer’s concerns (Lines 246-252).

3. The third-to-last paragraph (187-196) appears to confuse the tilt-translation ambiguity (the inability of a single inertial detector to disambiguate linear acceleration and rotation) with the basics of reflexive stabilization of posture and gaze. The VOR will happen whether the fish is tilted or linearly translated, as (presumably) would the tail movements. These two are not “unrelated:” on the contrary, evidence from other vertebrates suggest that they represent a collective attempt to correct destabilizations in both posture and gaze. As fish maintain a dorsal-up posture, if a fish were to roll (or be translated in the dark) to the left, it would be expected counter-roll its eyes to the right. Similarly, it ought do whatever it needed to its tail to ensure that it rolled to the right (more on this below). These have nothing to do with the nature of the stimulus that displaces the otolith. This confusion is also present in [33] where the authors separate acceleration from roll. There is no evidence that the fish perceives the two differentially, and the fact that the authors observe eye movements and tail movements do not disambiguate the two.

I think that the problem here lies in our presentation of the concept. We agree on all points. Either acceleration or rotation results in a similar movement of the otoliths relative to their surroundings, and therefore similar outputs from the hair cells, with resulting similar behaviour. We use the phrase “apparently unrelated” to set up an explanation of the tilt-translation ambiguity. We did this so that we could spend the rest of the paragraph explaining how this was not a paradox, and how these responses were, in fact, related. Apparently, this approach of was ineffective, given that it confused a expert in the field. We have scrapped this paragraph structure, and rewritten it in a more orthodox fashion. The reworked paragraph spans lines 217-232.

We have also changed “acceleration and roll” to “vestibular stimuli” in line 60.

4. The authors claim that the tail movements would “likely represent postural changes.” [193] I encourage the authors to expand on this. Specifically, the authors should offer a compelling physical basis for this interpretation. I encourage the authors to look at Ehrlich and Schoppik, 2017. That paper found that any forward translation was sufficient to correct pitch. There is no reason to suppose that this finding wouldn’t extend to roll tilts as well. Specifically, if a fish translates, as it does when it bouts, its torpedo-like morphology is inherently stabilizing. Such an interpretation is consistent with the supplemental video and in (Figure 3d), where stronger stimuli are accompanied by an increased probability of forward swims. N.B. I may have missed it but I don’t believe the authors explicitly describe what constitutes a “forward swim,” though I surmise it is what we see in the caudal tail in the video. If the authors really want to propose that it is the tail angle and not the instantiation of the swimming that would stabilize posture I strongly encourage them to propose a testable mechanism by which a yaw angle of the tail would serve to stabilize the roll axis of the fish. They might model the effects of a body angle given the constant angular acceleration in pitch to which larvae are subject (Ehrlich and Schoppik 2017); the combined pitch & yaw might be sufficient to roll the fish properly for small deviations from dorsal-up. Similarly, it would be good to know if the first deflection of the tail in a forward swim was oriented preferentially to the left/right depending on the otolith stimulus that generated it? Slight deviations in the strength of the beat might be similarly useful in stabilizing posture. Ultimately, larval zebrafish morphology is sufficient to ensure that simply generating a swim bout would stabilize roll, regardless of direction.

This is a very useful perspective. Thank you. We agree that our linking the tail movements to directional control, rather than roll correction, is an over interpretation of our data. The Ehrlich and Schoppik paper published since our submission, in fact, provides what is probably a more compelling explanation. Ultimately, we do not know enough to say exactly what the tail movement is intended to do (which we now state in lines 227-229). Instead, we have added a reference to the Ehrlich paper, and have added a discussion of how this paper’s outcomes impinge on the interpretation of our results (Lines 229-231).

5. Analysis of the eye movement data, while generally adequate for the claims that the authors make, is difficult to interpret given the authors’ choice of example movies. Specifically, the fish make spontaneous naso-temporal saccades and relax the eyes along this axis; this appears to be visible in the three-part video, particularly in the third part. If that particular video is comprised of three videos put together (as I’m guessing because of the displacements of the eye and the fish body), it would help to insert blank frames so the readers are not misled into thinking it contiguous. The authors would also do well to analyze any rostral-caudal changes in the pigment fiducials, and not just medio-lateral. A truly medio-lateral deflection of the utricle would produce no systematic rostral-caudal changes that would be the hallmark of a torsional (i.e. response to pitch

tilts) eye movement. I note this because the authors claim that [161] “OTs directed at the centres of the otoliths produced no responses (Fig 3j Movie S6) but repeated viewings of Movie S6 shows a clearly discernible movement of the fish’s right eye. Finally, while I’m loathe to suggest new experiments, there are clear predictions for the eye movements that force on the rostral or caudal part of the utricle should produce, not just the medial/lateral edge. The manuscript would be considerably stronger for the characterization of eye and tail responses to the other two directions to which the utricle is sensitive.

We have edited the relevant movie to make it clear that three separate segments of video are being shown. This has been done by adding black frames in between the separate clips. We have also analysed rostro-caudal movements across all movies, and failed to measure any movement. Panels a and b in Figure 4 have been changed to avoid misleading rotation angles. The new panels show the rotation of the eyes when the eyes are straight and not at an angle which would still lead to a roll only rotation but would look like a combination of rostro-caudal and medial-lateral movement.

6. The authors show the utricle moving on order 100um in Figure 1. I was surprised not to be able to see any deflection at all in Movies S5 and S6. Presumably this is because the otolith itself is tethered, but the authors might comment on this (unless I missed it somewhere)

We believe that we have addressed this question, raised by Reviewer 1 in his/her Comment 4.

7. The authors rely heavily on the “centre” directed stimulus to control for “heating, pain or damage” to the otolith. I note, though, that there is an extensive literature on laser-induced currents in hair cells: Rabbitt RD et. al. 2016, Rajguru 2011, and for shorter wavelength NIR stim Xia et. al. 2014 10.1109/EMBC.2014.6944163 . The manuscript would be stronger for a discussion of these results, as the “centre” stimulus does not adequately control for the possibility that the laser light itself could be sufficient to excite utricular hair cells, as shown by a number of other groups.

We agree that omitting these studies was an oversight. In these studies, however, IR radiation was applied directly to the hair cells, rather than the otoliths. This does not exclude the possibility that diffracted IR light could hit and stimulate the hair cells, but this should happen in the centre control as well as the trap at the edge of the otolith. In our view, the most compelling piece of evidence that our results are the products of physical forces on the otoliths is the opposite tail bend that occurs when a medial trap is released. The opposing effects of medial and lateral traps, and the nonresponse to traps in the centre, we believe, implicate forces on the otolith as the origins of our behavioural responses. We have now added a sentence citing these papers and mentioning that off-target stimulation of the hair cells was a confound that we needed to address with our control experiments. This can be found in lines 158-161.

[40] should read “sensitive **to** rotational”

Thank you. Corrected, now in line 28.

REVIEWERS' COMMENTS:

Reviewer #1 (Remarks to the Author):

In the response the authors answered in a satisfying manner to most of the issues raised in my previous report and revised the manuscript accordingly.

It is only issue 6 that was not fully addressed. The type of experiment I had in mind was, e.g., to use a fish mutant with an inactivated hair cells response or maybe, as also written by the authors, to use gene silencing. However, I still think the study is highly interesting and worthwhile publishing and I will be OK with publishing the manuscript in its current state even though issue 6 is not fully addressed.

Reviewer #2 (Remarks to the Author):

The authors have addressed all of my concerns. The final manuscript is strong, likely to be of broad interest. I recommend publication forthwith.

I would like to call the authors' attention to the following paper: [10.1006/dbio.1997.8736](https://doi.org/10.1006/dbio.1997.8736)
A Critical Period of Ear Development Controlled by Distinct Populations of Ciliated Cells in the Zebrafish by Riley et. al.

There, the authors use optical trapping to manipulate the developing otolith and test the role of different mutations & environmental conditions. The earlier work does not detract in any way from the novelty or scope of the manuscript under review, but given the similarity I encourage acknowledgement of this work.